# Does behavior mediate the effect of weather on SARS-CoV-2 transmission? evidence from cell-phone data

Elise N. Grover[1], Andrea G. Buchwald[2], Debashis Ghosh[3], Elizabeth J. Carlton[1]*

1 Department of Environmental and Occupational Health, Colorado School of Public Health, University of Colorado Anschutz Medical Campus, Aurora, Colorado, United States of America, 2 Center for Vaccine Development and Global Health, University of Maryland School of Medicine, Baltimore, Maryland, United States of America, 3 Department of Biostatistics & Informatics, Colorado School of Public Health, University of Colorado Anschutz Medical Campus, Aurora, Colorado, United States of America

* elizabeth.carlton@cuanschutz.edu

**Data Availability Statement:** The data used in this analysis cannot be shared publicly because it was collected and provided to the authors by third parties. The following third parties can be

## Abstract

There is growing evidence that weather alters SARS-CoV-2 transmission, but it remains unclear what drives the phenomenon. One prevailing hypothesis is that people spend more time indoors in cooler weather, leading to increased spread of SARS-CoV-2 related to time spent in confined spaces and close contact with others. However, the evidence in support of that hypothesis is limited and, at times, conflicting. We use a mediation framework, and combine daily weather, COVID-19 hospital surveillance, cellphone-based mobility data and building footprints to estimate the relationship between daily indoor and outdoor weather conditions, mobility, and COVID-19 hospitalizations. We quantify the direct health impacts of weather on COVID-19 hospitalizations and the indirect effects of weather via time spent indoors away-from-home on COVID-19 hospitalizations within five Colorado counties between March 4th 2020 and January 31st 2021. We also evaluated the evidence for sea-sonal effect modification by comparing the results of all-season (using season as a covari-ate) to season-stratified models. Four weather conditions were associated with both time spent indoors away-from-home and 12-day lagged COVID-19 hospital admissions in one or more season: high minimum temperature (all-season), low maximum temperature (spring), low minimum absolute humidity (winter), and high solar radiation (all-season & winter). In our mediation analyses, we found evidence that changes in 12-day lagged hospital admis-sions were primarily via the direct effects of weather conditions, rather than via indirect effects by which weather changes time spent indoors away-from-home. Our findings do not support the hypothesis that weather impacted SARS-CoV-2 transmission via changes in mobility patterns during the first year of the pandemic. Rather, weather appears to have impacted SARS-CoV-2 transmission primarily via mechanisms other than human move-ment. We recommend further analysis of this phenomenon to determine whether these find-ings generalize to current SARS-CoV-2 transmission dynamics, as well as other seasonal respiratory pathogens.

contacted to request access to the data used in this analysis: Colorado Department of Public Health and Environment (CDHPE), METADATA by John Abatzoglou, PurpleAir, Microsoft, and X-Mode. Data on reportable diseases in Colorado is made available upon request from CDPHE at https://www.datarequest.dphe.state.co.us/requests/create. The METADATA gridMET daily meteorological summary dataset is available for download at https://www.climatologylab.org/gridmet.html. PurpleAir data can be obtained from: https://api.purpleair.com/. Colorado Building Footprints data is made available by Microsoft here: https://github.com/microsoft/USBuildingFootprints. X-Mode data was obtained under a research use agreement by Vanadata.io, who also processed the data. Inquiries about access to X-Mode data can be made at: https://outlogic.io/contact/.

**Funding:** The work of ENG was supported through the MIDAS Coordination Center (MIDASSUP2020-3) by a grant from the National Institute of General Medical Science (3U24GM132013-02S2), Additionally, AGB and EJC were supported under Cooperative Agreement number NU38OT000297 from The Centers for Disease Control and Prevention (CDC) and the Council of State and Territorial Epidemiologists (CSTE). The work presented in this paper does not necessarily represent the views of CDC and CSTE. ENG, AGB, DG, and EJC all received funding from the Colorado Department of Public Health and Environment. The funders had no role in study design, data collection and analysis, decision to publish, or preparation of the manuscript.

**Competing interests:** The authors have declared that no competing interests exist.

## Introduction

Since the earliest days of the COVID-19 pandemic, there has been speculation regarding whether SARS-CoV-2 would follow the typical seasonal pattern of respiratory viruses seen in temperate climates, where transmission tends to increase in winter and wane during the summer [1]. Four years after the discovery of SARS-CoV-2, a growing body of evidence suggests that SARS-CoV-2 is impacted by weather, similar to what is observed for seasonal respiratory infections such as influenza [2–6]. Literature to date suggests evidence of increased SARS-CoV-2 transmission at cooler temperatures [7], and to a slightly lesser extent, at lower humidity [8]. However, it is less clear what drives these relationships.

A paradox in the seasonal transmission of respiratory illnesses is that year-round the average person spends the majority of their time indoors [9]. Additionally, SARS-CoV-2 transmission is primarily linked to indoor environments where crowding and lack of ventilation facilitate transmission. This begs the question: how are respiratory viruses such as SARS-CoV-2 linked to weather when people spend little time exposed to outdoor meteorological conditions and transmission primarily occurs in indoor, conditioned spaces? A common explanation is that weather alters human movement patterns, leading people to spend more or less time indoors, altering the frequency and length of time spent in close contact with others [10–13]. Here the ostensible relationship is that as weather becomes more unfavorable, people spend more time indoors and in indoor, confined spaces, increasing close contact with others and the probability of respiratory virus transmission. Despite its plausibility, the evidence for these phenomena is limited and conflicting. For example, while researchers in Belgium found a significant increase in long duration (>1 hour) social contact on low temperature work days as compared to high temperature days, they also found that the total number of social contacts that a person experienced during the workday did not significantly change with temperature [13]. Likewise, surveys of American and Canadian activity patterns have demonstrated that people tend to spend more time indoors during the winter than the summer [10], but a 2019 study from the US northwest demonstrated that snowfall-related school and workplace closures led to reduced social contact and subsequent reductions in cumulative respiratory virus incidence [14].

There is considerable evidence that human mobility impacts SARS-CoV-2 transmission, including evidence that human mobility patterns are a stronger driver of transmission than weather. Studies estimating the relative contribution of mobility-related indicators (e.g., mobility index, countrywide lockdown, homestay) versus various weather conditions have found that mobility indicators were stronger contributors to SARS-CoV-2 transmission and COVID-19 pandemic growth than any weather conditions on their own [15–17]. Similarly, two panel studies reporting the net effect of mobility and weather on SARS-CoV-2 transmission within countries with high confirmed COVID-19 cases during the first six months of the pandemic highlighted the potential suppression effect of mobility [18,19], which occurs in mediation analyses when the direct and indirect (i.e., mediated) effects have opposite signs [20]. That is, any reductions in SARS-CoV-2 transmission that may have resulted from the negative association between temperature (or solar radiation) and SARS-CoV-2 were ultimately counteracted by warm temperature-related rises in mobility and the subsequent rises in mobility-related SARS-CoV-2 transmission [18,19].

There are several plausible pathways by which weather can directly impact transmission outside of mobility. For example, studies have demonstrated that a host's innate immune response tends to be improved at higher temperatures [21], while lower temperature and humidity have been found to impair airway tissue repair and mucous production for capturing and transporting invaders [22–25]. Studies have also suggested that lower sun exposure and

subsequent lower levels of Vitamin D is associated with a decrease in host immunity and increased respiratory viruses during winter months [26,27]. Ambient temperature and humidity determine how quickly particles ejected by a cough or sneeze evaporate, which has the potential to affect the average distance travelled and the time that particles remain airborne [28]. While several factors are important to consider when it comes to the transmissibility of ejected particles–such as particle size, speed of ejection, ventilation, floor plan, etc.,–in general the effectiveness of airborne transmission is likely to increase when temperature and relative humidity are lower [28–30]. Furthermore, lower temperatures and humidity have been associated with greater viral viability, stability and survival [31,32]. One comparison of virus survival at room temperature versus summer temperatures found much more rapid decay at summer temperatures [33], while other studies have demonstrated that the SARS-CoV-2 deactivation rate related to sunlight and solar radiation is slower during the winter season [34,35].

We conducted a mediation analysis to test the hypothesis that meteorological conditions impact COVID-19 transmission via changes in human mobility patterns. We use daily COVID-19 hospital admissions in five Colorado counties between March 2020 and January 2021, daily weather data, building footprints, and cellphone-based mobility data to distinguish time spent indoors versus outdoors, and time spent at home versus away-from-home. Using the mediation analysis framework shown in Fig 1, we evaluate the extent to which weather directly influenced COVID-19 hospitalizations during the first year of the pandemic, as compared to an indirect mediation effect in which weather impacts mobility, which subsequently influences COVID-19 transmission. Instead of using absolute measures of weather, mobility and COVID-19 hospitalizations in this analysis, we standardize our weather variables and mean-center our mediators (i.e., mobility metrics) and outcome (i.e., COVID-19 hospitalizations) by county and season, such that the results could be interpreted as change relative to the county-season mean. This approach allowed us to account for season- and location-specific variability in our definitions of meaningful changes in weather, mobility and COVID-19 transmission. Additionally, we used publicly available indoor and outdoor climate data to evaluate the relationship between indoor and outdoor weather conditions. Through this analysis we test an often cited, yet largely suppositious hypothesis, with the goal of improving our understanding of the complex interplay of weather and behavior on a respiratory pathogen.

## Materials and methods

### Study period and population

We combine daily measures of indoor and outdoor weather conditions, mobility data from mobile devices, and COVID-19 hospitalization data for five Colorado counties between March 4th 2020 and January 31st 2021. March 4th was selected as our study's start date, as this corresponded with the first confirmed COVID-19 hospitalization in Colorado. We focus on Boulder, Denver, Douglas, El Paso and Mesa counties because of the availability of both indoor and outdoor weather and mobility data for the length of the study period. These counties all have populations above 150,000 people and are among the most populous in the state. Due to rapidly changing transmission dynamics related to increasing vaccinations, the rise of highly transmissible SARS-CoV-2 variants in the late winter of 2021, and an end to our access to the mobility data, we end the dataset on January 31, 2021. We analyze the data by season using the following definitions: Spring from March 4th to May 31st; Summer from June 1st to August 31st; Fall from September 1st to November 30th; and Winter from December 1st to January 31st

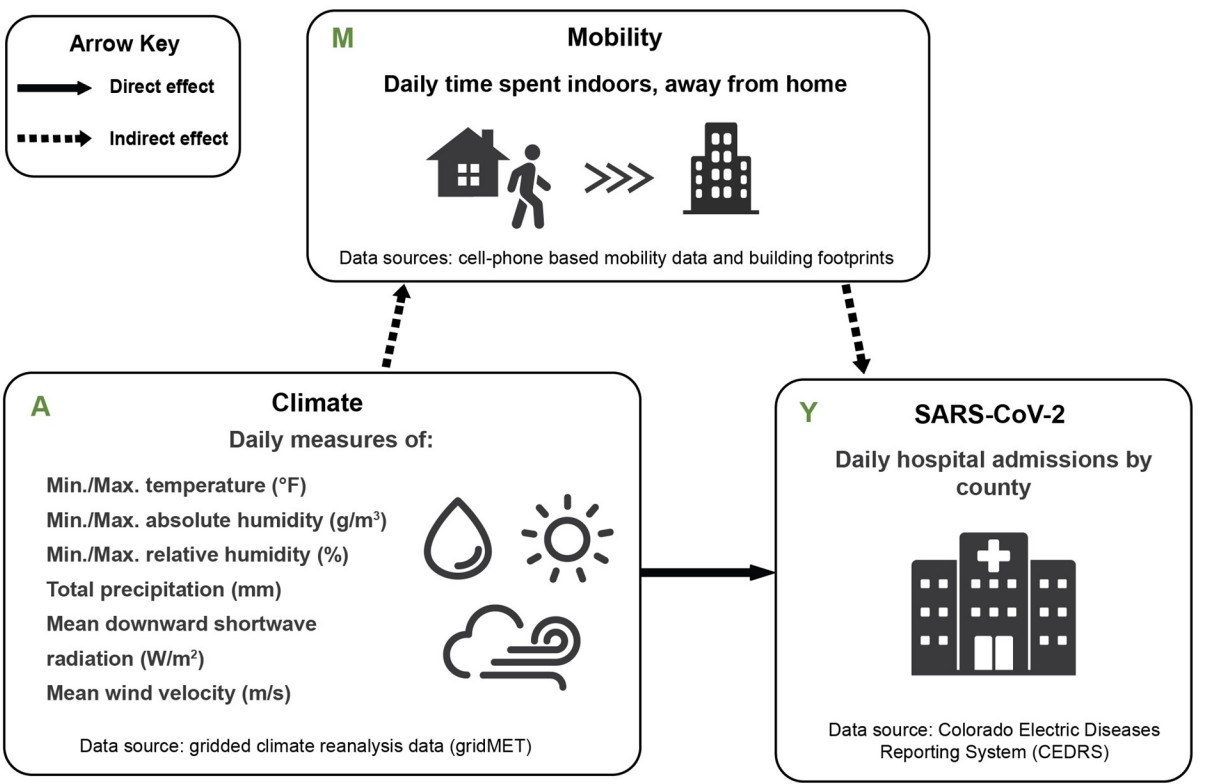

**Fig 1. Mediation analysis framework.** The hypothesized mechanisms by which outdoor weather can impact SARS-CoV-2. The indirect effect hypothesis is that weather changes human mobility patterns, specifically time spent in indoor spaces away from home where the virus can spread, leading to an increase in infections. An alternative, direct effect hypothesis is that weather directly impacts the spread of the virus by altering virus survival and transmission probability.

### Indoor vs. outdoor climate

We compile data from PurpleAir monitors (https://api.purpleair.com/) to assess the relationship between indoor and outdoor weather conditions at different times of year. PurpleAir monitors report temperature, humidity and air quality readings for monitors located both inside and outside. We generated estimates of the county-wide mean, standard deviation, 10th and 90th percentile values for indoor and outdoor temperature and humidity conditions each day using 10-minute interval readings from 1) all indoor monitors located in each county, and 2) all outdoor monitors located in each county. All monitors have two sensors that enable users to gauge the reliability of the readings. We omitted data when the two sensors were not within 10 units (degrees F and percent relative humidity) of each other or when the air quality reading of one sensor was at least three times greater than the other. We also omit all data from a single monitor if greater than 20% of its readings are outside of a plausible range, defined as greater than 100% or less than 0% for relative humidity, and temperatures higher or lower than the record high and low temperatures observed in Colorado for a given month. Notably, heat generated by the WiFi module of PurpleAir monitors causes them to be on average 8°F higher than the ambient temperature, and 4% lower than the ambient relative humidity [36].

## Climate data

We use gridMET data [37] to supplement daily outdoor weather conditions across each county because it is more accurate and reliable than the purple air sensors. GridMET is a gridded reanalysis dataset estimating daily surface meteorological conditions across the continental US at high spatial resolution (~4-km, 1/24th degree) that spans our study period [37]. We collect daily estimates of shortwave solar radiation (W/m$^2$), and the minimum and maximum temperature (˚F), relative humidity (%) and absolute humidity (g/m$^3$), as these are the weather variables that have been the most consistently associated with COVID-19 to date [7,8,38]. Additionally, we also evaluated daily precipitation (mm) and daily average windspeed (m/s), as some studies have highlighted these meteorological factors as being potentially associated with COVID-19, albeit with more variable findings than what's been found to date for solar radiation, temperature and humidity [39–46]. We calculate the average value of each weather metric on each day within a county to produce a county-day dataset.

As we were interested in relative (as opposed to absolute) changes in weather for a given location and season, each meteorological variable was standardized by county and season. To do this, the mean and standard deviation for each county-season was estimated. The county-season mean was then subtracted from each observation from a given county and season, and the resulting value was divided by the county-season standard deviation. Mean-centering by county and season allowed us to focus on location-specific variability in weather within each season, and facilitated a simple and uniform standard deviation-based definition of the "treatment" level of each weather variable for mediation analyses.

## Mobility data

We use mobile device data from X-mode (https://aws.amazon.com/marketplace/seller-profile?id=4e8835bd-89dc-4ae7-818d-52c22fedcbb9) consisting of timestamped geolocated points (i.e., device breadcrumbs). X-mode is one of many mobile device location data providers whose data have been used to study mobility patterns during the COVID-19 pandemic [47]. We use the mobile device data to estimate the average daily percentage of time that county residents spent indoors away-from-home. We use this measure as our primary description of transmission-relevant behavior because SARS-CoV-2 overwhelmingly occurs indoors [48,49], and, while the home environment is a key source of SARS-CoV-2 transmission, contacts outside of the home are necessary for introduction of the virus into the home [50]. Additionally, we conducted a sensitivity analysis using time at home as an alternate measure of transmission-relevant behavior, due to concern over potential misclassification that could arise when differentiating between time spent indoors versus outdoors.

Mobile device location was divided into five categories: indoors at home; outdoors at home; indoors not-at-home; outdoors not-at-home; and in transit. To determine if a mobile device was indoors or outdoors, we used a database of Colorado building footprints from Microsoft (https://github.com/microsoft/USBuildingFootprints). We classified the device as indoors if it met the following conditions: 1) the device remained within an 80-meter radius for at least 2 minutes; and 2) The geometric median of the pings within that radius rested within 1-meter of a building footprint. When a device was moving at a speed greater than 10 miles per hour, we classified that device as "in transit". We classified all other device activity–static, not within 1-meter of a building footprint, or moving less than 10 miles per hour–as outdoors. To determine when the device was away-from-home, we assigned two "home locations" per day per device: the "primary home"–the most common nighttime location for the device over the past 6 weeks–and the "current home"–the device's primary location from the prior night. We

considered a device to be away-from-home if it was observed more than 60-meters from both its primary and current home.

To generate daily estimates of where time was spent for each device, we expressed time spent in each of our five device location categories as a percentage, using the total amount of time a device was observed each day as the denominator. We then calculate a county-wide average of the percent of time spent in each of the five location categories for each study county. Finally, to facilitate comparisons across counties and seasons that account for intra-county and season variability in what constitutes "typical" mobility behaviors, we mean-centered each of our "percent of time spent" variables such that a value of "0" represented the county and season average percent of time spent in a given location, while a negative/positive number indicated lower/higher than average percent of time spent in that location, given the county and season.

## COVID-19 hospitalization data

We used reported case data from the Colorado Electronic Disease Reporting System (CEDRS) provided by the Colorado Department of Public Health and the Environment (CDPHE) to calculate the daily number of people admitted to the hospital with SARS-CoV-2 in each county. We used COVID-19 hospitalizations as our outcome instead of reported cases because we suspect cases reflect a variable proportion of infections over time due to changes in access to testing and test-seeking behaviors across our study period. The hospitalization data reflect COVID-19 hospitalizations from March 4th 2020 to January 31st 2021 reported to CEDRS accessed on January 3, 2022. Co-authors (EJC) had access to identifiable records that were used to generate the aggregated daily measures. To approximate a normal distribution and to facilitate interpretation across counties and seasons that account for intra- county and season variability in COVID-19 hospitalizations, we mean-centered hospitalizations for each county and season such that a value of "0" represented the average number of hospitalizations for a given county and season, while a negative/positive number indicated the degree to which daily hospitalizations were below/above the county and season average.

We hypothesized that meteorological conditions impact behavior immediately, leading to SARS-CoV-2 transmission events. COVID-19 hospitalizations, however, occur days after such transmission events. We lag hospitalizations by 12-days. This 12-day lag is the sum of a literature-based estimate of the average incubation period for SARS-CoV-2 of 4-days [51,52], plus our data-based average of 8-days between reported symptom onset and hospitalization for our study population and period, based on an analysis of the CEDRS data [53].

## Ethics

This project was determined to be exempt from Institutional Review Board review by the Colorado Multiple Institutional Review Board.

## Statistical analysis

**Indoor vs. outdoor weather.** First, to evaluate the relationship between indoor and outdoor weather conditions, we calculated Pearson's correlation coefficients between daily estimates of indoor and outdoor mean temperature, relative humidity and absolute humidity for each 3-month season derived from the Purple Air sensors, using the benchmark values outlined by Schober et al., (2018) to indicate strong (0.70–0.89) and very strong (0.90–1.00) correlation [54]. Scatter plots were also used to visualize the relationship between indoor and outdoor conditions across the study period.

**Mediation screening.** We then conducted a mediation analysis to evaluate evidence that meteorological variables impact COVID-19 hospitalizations directly or via changes in human mobility. Mediation analysis is recommended only when there is evidence of an association between the exposure and mediator, the exposure and outcome, and the mediator and outcome. We first screen each weather variable for evidence of an association with the mediator and with the outcome, using an approach that allowed us to consider a wide range of candidate weather variables, as described in the sections below.

**Nonlinear effects of weather.** Recognizing that the relationship between weather and behavior, and weather and COVID-19 may be non-linear, we converted each of our standardized meteorological variables into 3-category versions to capture "high", "middle" and "low" values across their distributions. This allowed us to avoid assumptions of linearity and to generate data-informed definitions of our treatment variables for the mediation analyses. Lowess plots were used to examine the shape of the relationship between weather and human mobility, as well as weather and COVID-19 hospitalizations [55]. As the default, we used ± 1 SD as cut points for generating 3-category weather variables (i.e., <-1 SD, -1–1 SD, >1 SD), though in some instances, alternative cut points were selected based on the relationships indicated via the Lowess plots.

**Weather vs. human mobility.** Once we determined the cut points to use for each categorical weather variable, we assessed its independent relationship with human mobility (i.e., time spent indoors, away-from-home) using linear regression models. For each linear regression model, we used Beta coefficients ($\beta$), a 95% confidence interval (CI) and a probability value (p-value) of $<0.05$ to indicate whether above (e.g., $> 1$ SD) or below average (e.g., $< -1$ SD) weather conditions (relative to the mid-range values of -1 to +1 SD) were significantly associated with county-season averaged percent of time spent indoors away-from-home.

Each of our linear regression models used time spent indoors away-from-home (mean-centered by county-season) as the outcome, included a single meteorological variable (to avoid issues with multicollinearity), an auto-correlation term (i.e., yesterday's value for time-spent indoors, away-from home) as well as three variables hypothesized to be sources of confounding: day type, stay-at-home orders, and rising Colorado hospitalizations last week. Day type was a binary variable generated to account for potential changes in behavior associated with different days of the week/month, indicating whether a given day was a weekday (Monday–Friday) versus a weekend (Saturday/Sunday) or a common 2020 US holiday (New Years Day, MLK, President's Day, Memorial Day, Independence Day, Labor Day, Columbus Day, Veteran's Day, Thanksgiving Day, Black Friday, Christmas Eve, Christmas Day, New Year's Eve). To account for potential shifts in behavior and hospitalizations during the early stage of the pandemic when COVID-19 lockdown mandates were in place in Colorado counties, we used a binary indicator of whether each county was under a stay-at-home order for a given day (March 26th–April 26th for Douglas, El Paso and Mesa counties; March 26th–May 8th for Boulder and Denver counties). Finally, to account for potential behavioral changes in response to perceived risk of COVID-19 during times of rising transmission, we compiled hospitalization census data (indicating total COVID-19 hospitalizations each day) to calculate an average weekly number of hospitalizations (avhosp) across the entire state of Colorado. This was used to generate a binary indicator of whether Colorado hospitalizations were increasing last week (hospgrowth), as compared to the week prior, calculated as follows:

$$hospgrow = 1 \; if \; \frac{(avhosp\_wk_{n-1} - avhosp\_wk_{n-2})}{avhosp\_wk_{n-1}} > 0, else \; hospgrow = 0$$

Recognizing that the direction and shape of the relationship between human mobility and weather could vary by season, we evaluated the evidence for seasonal effect modification by comparing the results of all-season (using season as a covariate) to season-stratified linear regression models. We considered there to be evidence of effect modification when the stratum specific coefficient estimates from our regression models changed directions, or shifted by more than ± 25% between any two stratum.

**Weather vs. COVID-19 hospitalizations.** We again used linear regression to evaluate evidence of an association between the same set of weather variables and COVID-19 hospital admissions. Here we modeled COVID-19 hospital admissions (mean-centered by county-season to approximate a normal distribution) as the outcome and included a single 3-category meteorological variable as the treatment/exposure variable, an auto-correlation term (i.e., yesterday's value for COVID-19 hospital admissions), as well as the aforementioned potential confounders: day type, stay-at-home orders, and rising Colorado hospitalizations in the previous week. For each linear regression model, we used β-coefficients (β), a 95% CI and a p-value of <0.05 to indicate whether above (e.g., > 1 SD) or below average (e.g., < -1 SD) weather conditions (as compared to mid-range values of -1 to +1 SD) were significantly associated with county-season averaged COVID-19 hospital admissions. Again, we evaluated the evidence of seasonal effect modification by comparing the results of all-season (using season as a covariate) to season-stratified linear regression models.

**Mediation analysis.** The results from the aforementioned linear regression models between weather and human behavior, and between weather and COVID-19 hospital admissions were used as a 2-step screening tool to determine which weather variables and "treatment" definitions to assess in the mediation analysis. We included only those weather variables whose comparison of "treatment" vs "control" (either high vs. middle, low vs. middle, or both) was significantly associated (p-value<0.05) with both time spent indoors away-from-home and COVID-19 hospital admissions in the mediation analysis. In the event that our regression models suggested a significant association at only one treatment level (e.g., high vs. middle, but not low vs. middle), only that treatment level was assessed in the mediation model. For those weather variables that passed the 2-step screening, we then applied a mediation framework to determine the degree to which the total effect on COVID-19 hospital admissions was attributable to indirect effects (i.e., via time spent indoors away-from-home), versus via direct effects (i.e., via the weather variable). For each mediation model, we used the "mediate" command available in Stata 18 [56]. We defined the "treatment" weather group as either the "low" or "high" category (depending on which category was highlighted as being significantly different from the "middle" category in our regression models), while our default control group was always the "middle" category. Each mediation model also included the auto-correlation terms and the three control variables from our linear regression models (weekend/holiday, stay- at-home order, perceived risk), as well as an additional interaction term to account for interaction between the mediator and the treatment variable in each model. The latter decision is based on the guidelines laid out by VanderWeele (2015), wherein researchers are advised to include the interaction term by default, removing it only if its inclusion does not substantially change effect estimates [57].

**Sensitivity analyses.** We included three sensitivity analyses in this assessment. First, we assessed percent of time spent at home (both indoors and outdoors) as the mediator and compared these results to that of the results from the primary assessment of time spent indoors away-from-home as the mediator. Second, we assessed each of our meteorological variables as continuous (instead of categorical) exposure/treatment variables (using ± 1 SD as the "treatment" group in the mediation analysis, and 0 as the "control" group), and compared the results with those from the primary assessment. Third, we also reran our mediation models to exclude

the "hospgrowth" variable and compared the results with those models that included them to determine whether its inclusion resulted in any substantial changes in our results. Stata 18 was used for all analyses.

## Results

Between March 4th 2020 and January 31st 2021, there were a total of 8,827 COVID-19 hospital admissions within our five Colorado study counties (Table 1). The number of daily COVID-19 hospital admissions reported by each county was highest in the winter (Mean = 9.0, Standard Deviation (SD) = 7.6), and lowest in the summer (Mean = 1.7, SD = 2.2). A plot of COVID-19 hospitalizations within each county during our study period is available in Fig 2.

On average, people spent the most time away-from-home in indoor spaces during the fall (mean = 5.6%, SD = 1.2), and the most time at home during the spring (mean = 77.1%, SD = 5.0). Notably, the spring season corresponded with the executive "stay-at-home" order, which lasted from March 26th–April 26th 2020 for all Colorado counties and was extended by Denver and Boulder counties through May 8th.

The weather within our five study counties was characteristic of temperate and dry climate types across the study period (Table 1). The mean maximum daily temperature was highest in the summer (mean maximum temperature = 85.2, SD = 6.9) and lowest in the winter (mean maximum temperature = 41.7, SD = 10.1), with the largest variance experienced during the spring (mean maximum temperature = 61.4, SD = 13.5) and fall (mean maximum temperature = 63.9, SD = 16.8). Relative humidity tended to be lowest during the summer (mean

**Table 1. Summary of hospitalizations, behavior and weather for five Colorado counties between March 4th 2020 and January 31st 2021.**

|  | All-season | Spring | Summer | Fall | Winter |
|---|---|---|---|---|---|
| **Five county data summary** | N | N | N | N | N |
| Total observations (county-days) | 1670 | 445 | 460 | 455 | 310 |
| Total COVID-19 hospital admissions | 8827 | 1956 | 775 | 3307 | 2789 |
| **Hospitalizations** | Mean (SD) | Mean (SD) | Mean (SD) | Mean (SD) | Mean (SD) |
| Average daily hospital admissions, by county | 5.3 (7.3) | 4.4 (6.7) | 1.7 (2.2) | 7.3 (9.1) | 9.0 (7.6) |
| **Mobility** | Mean (SD) | Mean (SD) | Mean (SD) | Mean (SD) | Mean (SD) |
| Mean percent of time indoors, away-from-home (%) | 5.2 (1.6) | 4.3 (2.2) | 5.4 (0.9) | 5.6 (1.2) | 5.3 (1.1) |
| Mean percent of time outdoors, away-from-home (%) | 7.9 (1.5) | 7.1 (1.8) | 9.1 (0.6) | 8.2 (1.2) | 7.1 (0.9) |
| Mean percent of time indoors, at home (%) | 48.6 (3.0) | 50.0 (3.9) | 47.9 (2.2) | 48.3 (2.7) | 48.3 (2.2) |
| Mean percent of time outdoors, at home (%) | 25.5 (2.4) | 27.1 (2.7) | 23.9 (1.5) | 24.8 (1.7) | 26.3 (1.8) |
| Mean percent of time spent in transit (%) | 6.8 (1.4) | 5.7 (1.5) | 7.9 (0.7) | 6.9 (1.1) | 6.5 (0.7) |
| **Outdoor weather** | Mean (SD) | Mean (SD) | Mean (SD) | Mean (SD) | Mean (SD) |
| Minimum daily temperature (˚F) | 36.8 (15.5) | 34.2 (9.8) | 54.4 (5.5) | 34.4 (12.3) | 17.8 (6.4) |
| Maximum daily temperature (˚F) | 65.0 (19.3) | 61.4 (13.5) | 85.2 (6.9) | 63.9 (16.8) | 41.7 (10.1) |
| Minimum daily relative humidity (%) | 22.8 (15.9) | 25.3 (15.1) | 16.0 (8.5) | 19.8 (16.6) | 34.1 (17.6) |
| Maximum daily relative humidity (%) | 65.0 (19.4) | 69.0 (17.3) | 57.8 (17.2) | 61.4 (20.0) | 74.9 (18.9) |
| Minimum daily absolute humidity (g/m³) | 1.2 (0.7) | 1.3 (0.6) | 1.7 (0.9) | 0.9 (0.6) | 0.9 (0.4) |
| Maximum daily absolute humidity (g/m³) | 10.8 (5.6) | 9.9 (4.1) | 17.0 (4.7) | 9.3 (3.8) | 5.2 (1.6) |
| Mean downward shortwave radiation (W/m²) | 207.5 (78.6) | 242.5 (52.7) | 284.9 (30.4) | 168.3 (47.6) | 100.1 (13.8) |
| Mean wind velocity (m/s) | 8.3 (3.1) | 8.5 (2.5) | 7.7 (2.8) | 8.5 (3.5) | 8.8 (3.8) |
| Total daily precipitation (mm) | 0.9 (2.7) | 1.3 (3.9) | 1.0 (2.1) | 0.6 (2.3) | 0.5 (1.2) |

N = Number.
SD = Standard Deviation.

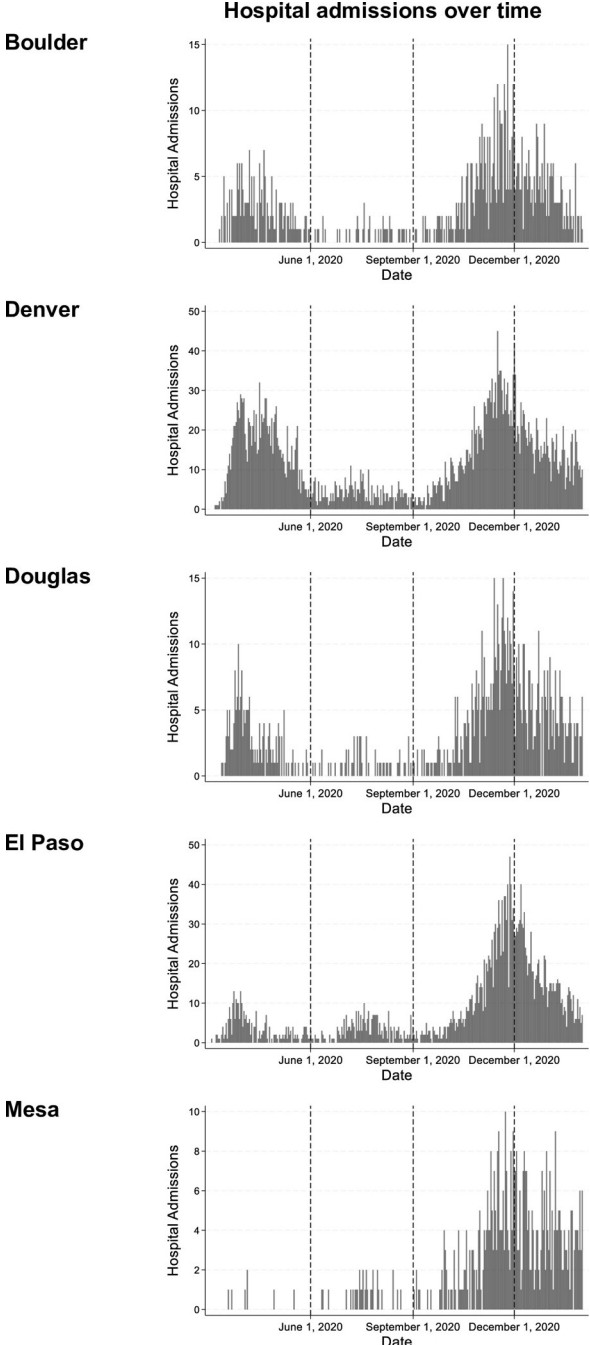

**Fig 2. Daily COVID-19 hospitalizations between March 4ᵗʰ 2020 and January 31ˢᵗ 2021 by county.** Comparison of daily COVID-19 hospitalization admissions between five Colorado counties between March 4ᵗʰ 2020 and January 31ˢᵗ 2021.

minimum relative humidity = 16.0, SD = 8.5; mean maximum relative humidity = 57.8, SD = 17.2) and highest during the winter (mean minimum relative humidity = 34.1, SD = 17.6; mean maximum relative humidity = 74.9, SD = 18.9), while absolute humidity was highest during the summer (mean minimum absolute humidity = 1.7, SD = 0.9; mean

maximum absolute humidity = 17.0, SD = 4.7) and lowest during the winter (mean minimum absolute humidity = 0.9, SD = 0.4; mean maximum absolute humidity = 5.2, SD = 1.6).

## Comparing indoor and outdoor weather conditions

A comparison of indoor and outdoor environmental conditions indicated that indoor conditions overlapped most with outdoor conditions between late spring and mid-fall (Fig 3). Indoor temperature and relative humidity were relatively stable across the year, though indoor temperature deviated from outdoor temperature between late fall and early spring (Fig 3, Panels A and B). Indoor absolute humidity showed greater variability and was closely correlated with outdoor absolute humidity (Fig 3, Panel C).

Correlation matrices comparing indoor and outdoor PurpleAir data demonstrate that the indoor and outdoor measures for mean temperature and absolute humidity were more strongly correlated than relative humidity across our study period (Fig 4). Correlation between indoor and outdoor temperature, relative humidity and absolute humidity was lowest during the winter season.

Correlation matrices showing Pearson's correlation coefficients and the strength of correlation between indoor and outdoor weather conditions between March 4th 2020 –January 31st 2021, and by season for five Colorado counties. Stronger correlation between outdoor and indoor measures are shown with darker tones.

## Weather vs. Mobility & Weather vs. COVID-19 hospitalization

The results of our linear regression models suggested that both above and below average weather conditions were frequently associated with the county-season average amount of time spent indoors away-from-home, but less frequently associated with 12-day lagged hospital admissions (Table 2). For example, when compared to mid-ranged minimum temperature, lower than average minimum temperature ($< -1$ SD) was significantly associated with an increase in the county-season mean percent of time spent indoors away-from-home in the all-season model ($\beta = 0.09$, 95% CI: 0.01, 0.17) and the spring model ($\beta = 0.30$, 95% CI: 0.11, 0.48), and a significant decrease in time spent indoors away-from-home in the winter model ($\beta = -0.32$, 95 CI: -0.47, -0.17) and the summer model ($\beta = -0.08$, 95% CI: -0.16, -0.00). Meanwhile, lower than average minimum temperature was not significantly associated with 12-day lagged COVID-19 hospital admissions in any of the season-stratified or all-season linear regression models.

Only four weather conditions were associated with both time spent indoors away-from-home and 12-day lagged COVID-19 hospital admissions in one or more season: high minimum temperature (all-season), low maximum temperature (spring), low minimum absolute humidity (winter), and high solar radiation (all-season & winter), which are highlighted in gray in Table 2. As our ultimate interests in the linear regression models were to 1) identify candidate weather variables (and their associated seasons) that were appropriately suited to a mediation analysis (i.e., those that were associated with both time spent indoors away-from-home and 12-day lagged COVID-19 hospital admissions), and 2) to determine the anticipated direction of association for those that met the conditions of mediation analysis. Table 2 includes only those regression results pertaining to the subset of weather variables that met those conditions. For further details on the association between time spent indoors away-from-home or COVID-19 hospital admissions and other weather variables (e.g., relative humidity, precipitation, wind speed) or the control variables, see S1 Table.

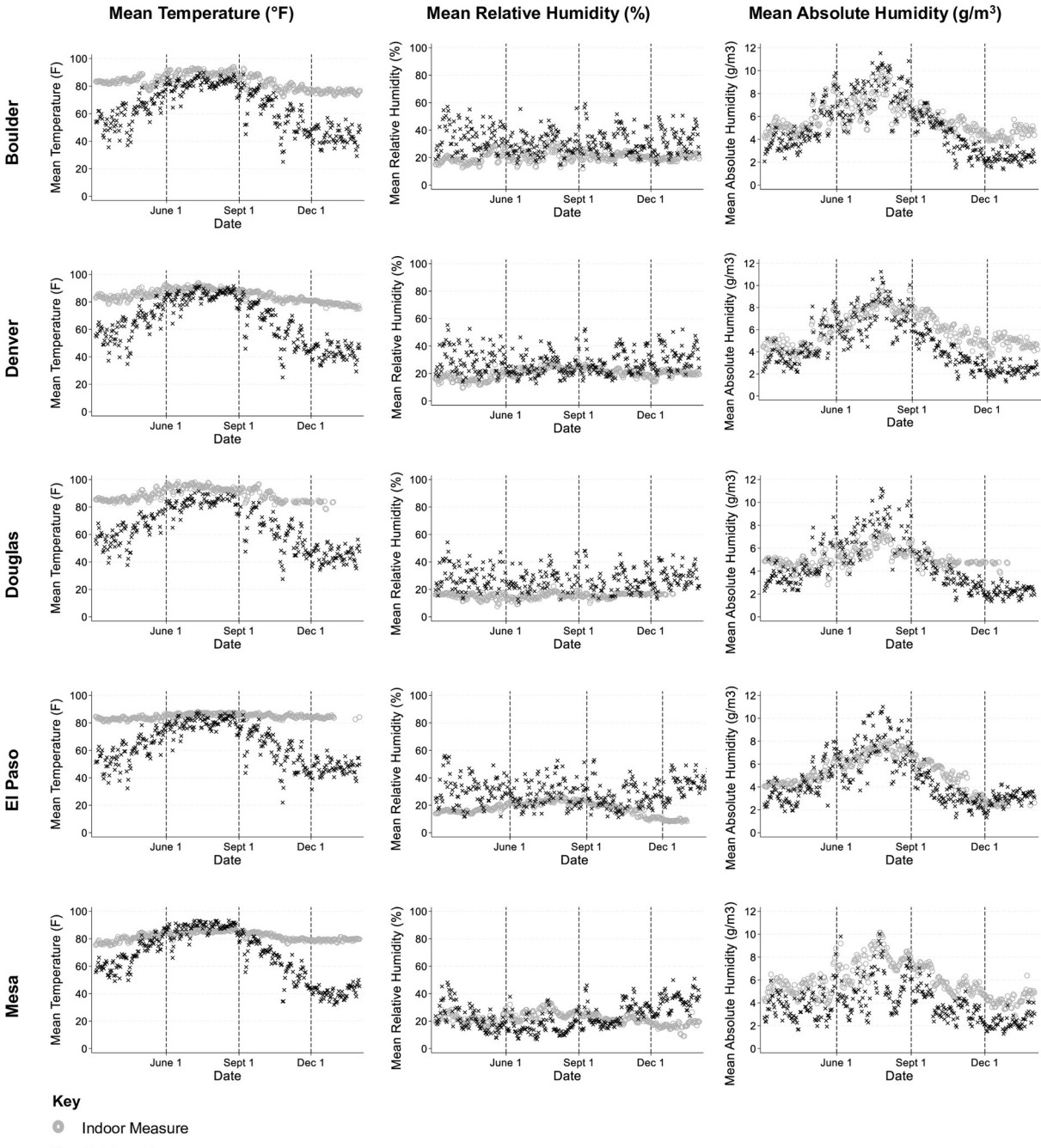

**Key**

○  Indoor Measure

X  Outdoor Measure

**Fig 3. Comparing indoor and outdoor temperature and humidity for five counties using data from geolocated sensors.** Scatter plots comparing indoor environmental conditions (blue circle) to outdoor environmental conditions (red triangle) between March 4th 2020 and January 31st, 2021 in five Colorado counties obtained from Purple Air. Vertical dotted lines indicate where one season ends and the next begins.

## Mediation analysis

In our all-season mediation models, change in the county-season average 12-day lagged COVID-19 hospital admissions was largely due to mechanisms other than the indirect route of time spent indoors away-from-home (Table 3). When minimum temperature was above

| | All-seasons | Winter | Spring | Summer | Fall |
|---|---|---|---|---|---|
| **Mean Temperature** | | | | | |
| All county | 0.7497 | 0.2512 | 0.4387 | 0.2113 | 0.6610 |
| Boulder | 0.8973 | 0.7252 | 0.3848 | 0.8460 | 0.9133 |
| Denver | 0.9282 | 0.2347 | 0.8318 | 0.5562 | 0.9117 |
| Douglas | 0.9067 | 0.0179 | 0.8312 | 0.5504 | 0.8448 |
| El Paso | 0.7899 | 0.6901 | 0.7991 | 0.7358 | 0.8086 |
| Mesa | 0.8541 | 0.1507 | 0.9449 | 0.7877 | 0.8769 |
| | | | | | |
| **Mean Relative Humidity** | | | | | |
| All county | 0.0821 | 0.0743 | 0.2898 | 0.2172 | 0.1561 |
| Boulder | 0.5109 | 0.3276 | 0.4041 | 0.9111 | 0.6474 |
| Denver | 0.1808 | 0.4456 | 0.5431 | 0.6728 | 0.0844 |
| Douglas | 0.5263 | 0.2441 | 0.5023 | 0.6679 | 0.5238 |
| El Paso | -0.0021 | -0.0267 | 0.1494 | 0.6678 | -0.0207 |
| Mesa | -0.0976 | 0.2133 | 0.8493 | 0.6985 | 0.1470 |
| | | | | | |
| **Mean Absolute Humidity** | | | | | |
| All county | 0.7889 | -0.0185 | 0.7679 | 0.5265 | 0.7292 |
| Boulder | 0.9369 | 0.2296 | 0.9259 | 0.9676 | 0.8413 |
| Denver | 0.8652 | 0.4299 | 0.8667 | 0.8320 | 0.9395 |
| Douglas | 0.8056 | 0.0771 | 0.4791 | 0.8059 | 0.6464 |
| El Paso | 0.8307 | 0.2377 | 0.8759 | 0.7578 | 0.7849 |
| Mesa | 0.8823 | 0.5470 | 0.9276 | 0.8528 | 0.8532 |

**Key**

| | |
|---|---|
| 0.00 – 0.10 | Negligible correlation |
| 0.10 – 0.39 | Weak correlation |
| 0.40 – 0.69 | Moderate correlation |
| 0.70 – 0.89 | Strong correlation |
| 0.90 – 1.00 | Very strong correlation |

**Fig 4. Correlation between indoor and outdoor environmental conditions by county and season.**

average ($> 0.5$ SD) in the all-season model, there was a small, but statistically non-significant increase in the average COVID-19 hospital admissions that was attributable to the indirect route of weather via time spent indoors away-from-home ($\beta = 0.02$, 95% CI: -0.01, 0.04). By contrast, the estimated direct effect of above average minimum temperature on average COVID-19 hospital admissions by county-season was an average decrease of 0.46 COVID-19 hospital admissions (95% CI: -0.82, -0.11). Thus, the resulting total effect of higher-than-average minimum temperature and time spent indoors away-from-home in the all-season model was an average decrease of 0.45 hospital admissions (95% CI: -0.80, 0.10) by county-season. In other words, there was a suppression effect, meaning the estimated impact of temperature-associated changes in mobility on COVID-19 hospitalizations was in the opposite direction of the estimated direct effect of temperature on COVID-19 hospitalizations. In a similar vein,

**Table 2. Relationship between meteorological variables and time spent indoors, away-from-home and COVID-19 hospital admissions estimated using linear regression.**

| Categorical weather conditions [a] | Daily county-mean percent of time spent indoors, away-from-home | | | 12-day lagged hospitalization admissions (mean-centered by county-season) | | |
|---|---|---|---|---|---|---|
| | β | 95% CI | p-value | β | 95% CI | p-value |
| **Minimum temperature** | | | | | | |
| **All-season [b]** | | | | | | |
| <-0.5 vs. mid | 0.09 | 0.01–0.17 | 0.028* | 0.14 | -0.25–0.53 | 0.490 |
| >0.5 vs. mid | -0.09 | -0.16 - -0.01 | 0.020* [c] | -0.41 | -0.78 - -0.03 | 0.034* [c] |
| Spring | | | | | | |
| <-0.5 vs. mid | 0.30 | 0.11–0.48 | 0.002* | -0.18 | -0.78–0.42 | 0.557 |
| >0.5 vs. mid | -0.19 | -0.41–0.04 | 0.099 | -0.23 | -0.96–0.49 | 0.526 |
| Summer | | | | | | |
| <-0.5 vs. mid | -0.08 | -0.16 - -0.00 | 0.039* | -0.35 | -0.77–0.07 | 0.102 |
| >0.5 vs. mid | -0.09 | -0.16 - -0.03 | 0.004* | -0.14 | -0.47–0.19 | 0.394 |
| Fall | | | | | | |
| <-1 vs. mid | -0.05 | -0.18–0.08 | 0.452 | 0.61 | -0.56–1.78 | 0.306 |
| >1 vs. mid | -0.08 | -0.20–0.04 | 0.210 | -0.03 | -1.12–1.06 | 0.959 |
| Winter | | | | | | |
| <-1 vs. mid | -0.32 | -0.47 - -0.17 | <0.001* | 0.26 | -0.69–1.21 | 0.589 |
| >1 vs. mid | 0.04 | -0.13–0.22 | 0.640 | 1.19 | -0.01–2.40 | 0.053 |
| **Maximum temperature** | | | | | | |
| **All-season [b]** | | | | | | |
| <-1 vs. mid | 0.08 | -0.00–0.17 | 0.060 | -0.19 | -0.62–0.23 | 0.374 |
| >1 vs. mid | -0.11 | -0.20–0.01 | 0.024* | -0.28 | -0.74–0.19 | 0.241 |
| Spring | | | | | | |
| <-1 vs mid | 0.23 | 0.03–0.44 | 0.028* [c] | -0.73 | -1.41 - -0.04 | 0.037* [c] |
| >1 vs mid | -0.12 | -0.42–0.18 | 0.448 | -0.62 | -1.58–0.33 | 0.201 |
| Summer | | | | | | |
| <-1 vs. mid | -0.05 | -0.14–0.04 | 0.303 | -0.48 | -0.94 - -0.02 | 0.039* |
| >1 vs. mid | -0.08 | -0.16 - -0.00 | 0.954 | -0.18 | -0.59–0.23 | 0.390 |
| Fall | | | | | | |
| <-1 vs. mid | -0.09 | -0.21–0.03 | 0.145 | 0.32 | -0.78–1.41 | 0.567 |
| >1 vs. mid | -0.13 | -0.26 - -0.01 | 0.041* | -0.28 | -1.42–0.86 | 0.632 |
| Winter | | | | | | |
| <-1 vs. mid | -0.23 | -0.37 - -0.08 | 0.003* | 0.40 | -0.59–1.39 | 0.431 |
| >1 vs. mid | 0.13 | -0.03–0.30 | 0.120 | 0.23 | -0.91–1.37 | 0.691 |
| **Minimum absolute humidity** | | | | | | |
| **All-season [b]** | | | | | | |
| <-1 vs. mid | 0.14 | 0.05–0.24 | 0.004* | 0.08 | -0.40–0.56 | 0.740 |
| >1 vs. mid | -0.09 | -0.18–0.00 | 0.053* | -0.31 | -0.77–0.15 | 0.183 |
| Spring | | | | | | |
| <-1 vs mid | -0.03 | -0.26–0.20 | 0.813 | 0.45 | -0.33–1.22 | 0.259 |
| >1 vs mid | -0.38 | -0.63 - -0.13 | 0.003* | 0.07 | -0.76–0.90 | 0.865 |
| Summer | | | | | | |
| <-1 vs. mid | 0.07 | -0.02–0.15 | 0.133 | -0.03 | -0.48–0.41 | 0.892 |
| >1 vs. mid | -0.10 | -0.17 - -0.02 | 0.011* | -0.16 | -0.55–0.22 | 0.401 |
| Fall | | | | | | |
| <-1 vs. mid | 0.18 | 0.04–0.33 | 0.014* | 0.68 | -0.66–2.01 | 0.320 |
| >1 vs. mid | 0.07 | -0.07–0.20 | 0.349 | 0.07 | -1.14–1.29 | 0.910 |

*(Continued)*

**Table 2.** (Continued)

| Categorical weather conditions [a] | Daily county-mean percent of time spent indoors, away-from-home | | | 12-day lagged hospitalization admissions (mean-centered by county-season) | | |
|---|---|---|---|---|---|---|
| | β | 95% CI | p-value | β | 95% CI | p-value |
| Winter | | | | | | |
| <-1 vs. mid | 0.22 | 0.03–0.40 | 0.017* [c] | -1.28 | -2.50 - -0.07 | 0.039* [c] |
| >1 vs. mid | -0.12 | -0.32–0.07 | 0.209 | -0.43 | -1.70–0.84 | 0.509 |
| **Solar Radiation** | | | | | | |
| All-season [b] | | | | | | |
| <-1.5 vs. mid | 0.02 | -0.11–0.14 | 0.798 | -0.77 | -1.41 - -0.14 | 0.017* |
| >0 vs. mid | -0.12 | -0.19 - -0.05 | 0.001* [c] | -0.74 | -1.09 - -0.39 | <0.001* [c] |
| Spring | | | | | | |
| <-0.5 vs mid | 0.39 | 0.16–0.62 | 0.001* | -0.49 | -1.20–0.23 | 0.183 |
| >0.5 vs mid | -0.37 | -0.62 - -0.11 | 0.004* | -0.26 | -1.09–0.58 | 0.544 |
| Summer | | | | | | |
| <-1 vs. mid | -0.07 | -0.16–0.02 | 0.129 | 0.03 | -0.47–0.54 | 0.903 |
| >1 vs. mid | -0.21 | -0.27 - -0.14 | <0.001* | -0.14 | -0.48–0.19 | 0.396 |
| Fall | | | | | | |
| <-1.5 vs. mid | 0.03 | -0.16–0.21 | 0.793 | -1.24 | -2.92–0.43 | 0.146 |
| >0 vs. mid | -0.01 | -0.10–0.08 | 0.795 | -1.49 | -2.39 - -0.61 | 0.001* |
| Winter | | | | | | |
| <-1.5 vs. mid | -0.11 | -0.31–0.10 | 0.317 | -0.78 | -2.18–0.62 | 0.276 |
| >0 vs. mid | 0.26 | 0.13–0.39 | <0.001* [c] | -1.07 | -1.94 - -0.20 | 0.016* [c] |

β = Beta coefficient.

CI = Confident Interval.

* p-value < 0.05.

[a] The beta coefficient, 95% confidence interval and p-value presented for each independent weather variable correspond with the adjusted models assessing the impact of each treatment variable on the mean percent of time indoors away-from-home (left) and 12-day lagged COVID-19 hospital admissions (right), controlling for holidays and weekends, the stay-at-home order, increasing Colorado hospitalizations, as well as an auto-correlation term indicating yesterday's response variable's value.

[b] Models that were not stratified by season instead included season as a covariate to account for season as a confounder.

[c] Indicates weather variables that were associated with both mean percent of time spent indoors away-from-home and 12-day lagged hospitalizations, as these criteria was used to determine which variables to assess in the mediation analysis.

when compared to mid-range solar radiation (-1.5 SD– 0 SD) in the all-season model, above average solar radiation (>0 SD) was associated with a small but statistically non-significant increase in COVID-19 hospital admissions via the indirect route of the associated changes in the average time spent indoors away-from-home (β = 0.03, 95% CI: -0.00, 0.07), but a significant decrease in COVID-19 hospital admissions via the direct route of higher than average solar radiation (β = -0.88, 95% CI: -1.23, -0.53).

Similar to the estimates from the all-season models, our season-stratified spring model suggested opposing effects of low maximum temperature and time spent indoors away-from-home: we found a non-significant increase in hospital admissions via the indirect route of time spent indoors away-from-home, but a statistically significant decrease in county-season average COVID-19 hospital admissions due to other mechanisms associated with lower than average maximum temperatures in the spring. In our two season-stratified winter models of the effects of low minimum absolute humidity and high solar radiation, we found that both the indirect effects and direct effects of these weather conditions each had a negative effect (albeit

**Table 3. Models assessing time indoors away-from-home as mediators between weather conditions and COVID-19 hospital admissions.**

| | Treatment level [a] | Effect | Estimate of the mediating effects of time indoors, away-from-home on 12-day lagged COVID-19 hospital admissions | | |
| | | | β | 95% CI | P-Value |
|---|---|---|---|---|---|
| **All-seasons** | | | | | |
| High minimum temperature | >0.5 SD vs. -0.5–0.5 SD | Natural Indirect Effect | 0.02 | -0.01–0.04 | 0.144 |
| | >0.5 SD vs. -0.5–0.5 SD | Natural Direct Effect | -0.46 | -0.82 – -0.11 | 0.011 |
| | >0.5 SD vs. -0.5–0.5 SD | Total Effect | -0.45 | -0.80 – -0.10 | 0.013 |
| High solar radiation | >0 SD vs. -1.5–0 SD | Natural Indirect Effect | 0.03 | -0.00–0.07 | 0.054 |
| | >0 SD vs. -1.5–0 SD | Natural Direct Effect | -0.88 | -1.23 - -0.53 | <0.001 |
| | >0 SD vs. -1.5–0 SD | Total Effect | -0.85 | -1.19 - -0.51 | <0.001 |
| **Spring** | | | | | |
| Low maximum temperature | <-1 SD vs. -1–1 SD | Natural Indirect Effect | 0.01 | -0.03–0.05 | 0.542 |
| | <-1 SD vs. -1–1 SD | Natural Direct Effect | -0.78 | -1.42 - -0.13 | 0.018 |
| | <-1 SD vs. -1–1 SD | Total Effect | -0.76 | -1.39 - -0.14 | 0.017 |
| **Winter** | | | | | |
| Low minimum absolute humidity | <-1 SD vs. -1–1 SD | Natural Indirect Effect | -0.10 | -0.35–0.16 | 0.462 |
| | <-1 SD vs. -1–1 SD | Natural Direct Effect | -1.19 | -2.44–0.06 | 0.063 |
| | <-1 SD vs. -1–1 SD | Total Effect | -1.29 | -2.59–0.01 | 0.052 |
| High solar radiation | >0 SD vs. -1.5–0 SD | Natural Indirect Effect | -0.09 | -0.26–0.08 | 0.300 |
| | >0 SD vs. -1.5–0 SD | Natural Direct Effect | -0.95 | -1.82 - -0.09 | 0.030 |
| | >0 SD vs. -1.5–0 SD | Total Effect | -1.04 | -1.87 - -0.21 | 0.014 |

β = Beta Coefficient.

CI = Confident Interval.

[a] Seasonal weather conditions were categorized into three groups by examining Lowess plots between the weather variable and both the mediator (time indoors away-from-home) and outcome (12-day lagged hospital admissions) in this analysis. Linear regression analyses compared the association of "high" and "low" weather categories (versus the mid-range) on both the mediator and outcome. Those seasonal weather conditions were significantly associated with both are included in this table.

statistically non-significant for all but the direct effects of high solar radiation) on county-season averaged COVID-19 hospital admissions.

In several instances, our indirect effect estimates yielded results that ran counter to the hypothesis that the indirect effect of spending more time indoors away-from-home would be an increase in COVID-19 hospital admissions (and vice versa). For example, in the case of higher-than-average minimum temperature in our all-season models, the linear regression models (Table 2) suggested that above average minimum temperature (>0.5 SD) was associated with a significant decrease in the county-season average percent of time spent indoors away-from-home (β = -0.09, 95 CI: -0.16, -0.01). By contrast, our mediation model highlighted a small increase (albeit statistically non-significant) in average COVID-19 hospitalizations attributed to temperature-induced changes in time spent indoors away-from-home, demonstrating a direct conflict with our hypothesis that a decrease in time spent indoors away-from-home would result in decreased risk of COVID-19 hospitalization.

## Sensitivity analyses

In a sensitivity analysis, we re-ran the analysis using time at home (either indoors or outdoors) as the mediator, hypothesizing that seasonal weather conditions would influence the amount of time spent at home, and that any time spent at home would be protective against COVID-

19. Compared to the linear regression results from the primary analysis, the results from this sensitivity analysis suggested that seasonal weather conditions were more strongly associated with time spent at home than with time spent indoors away-from-home (S2 Table). The mediation analysis showed similar results to that of our primary mediation analysis, demonstrating that changes in the county-season average 12-day lagged COVID-19 hospital admissions were primarily a result of mechanisms other than changes in time spent at home (S3 Table).

In an additional sensitivity analysis, we re-ran our linear regression models using continuous versions of weather variables. In our assessments concerning time indoors away-from-home, none of the continuous weather variables in the season-stratified models were statistically significantly associated with both the county-average percent of time spent indoors away-from-home and 12-day lagged COVID-19 hospital admissions (S4 Table). Among the all-season models, only minimum temperature was statistically significantly associated with both the county-average percent of time spent indoors away-from-home and 12-day lagged COVID-19 hospital admissions. The mediation analysis assessing the direct effects of continuous maximum temperature and the indirect effect of continuous maximum temperature via time spent indoors away-from-home yielded nearly identical results to that of the primary analysis (S5 Table). Similarly, when we repeated this analysis using continuous versions of the weather variables on time at home and COVID-19 hospital admissions, we again found that only the all-season minimum temperature was statistically significantly associated with both the county-average percent of time at home and 12-day lagged COVID-19 hospital admissions (S6 and S7 Tables).

In a final sensitivity analysis, we compared two sets of mediation models: those that included the "hospgrowth" variable to account for potential behavioral changes in response to perceived risk of COVID-19 during times of rising transmission, and those that excluded it. The results did not change substantially with the exclusion of the hospitalization growth variable for either the time spent indoors away-from-home (S8 Table) or time spent at home (S9 Table) mediation models.

## Discussion

In our analysis, we did not find evidence that weather impacted COVID-19 hospital admissions via changes in time spent indoors, away-from-home during the first year of the pandemic. Using a mediation framework, we found instead that the direct effects of weather on hospital admissions were far more influential than the indirect route whereby weather changes mobility behaviors. This has important implications, as it runs counter to the popular belief that seasonality in winter-dominant respiratory viruses is primarily a result of increased time spent indoors. We found minimal variability in time spent indoors away-from-home across seasons. Within and across seasons, we found that changes in time spent indoors away-from-home associated with weather had little to no effect on county-season averaged COVID-19 hospital admissions. By contrast, we found that relative changes in weather–specifically, above average minimum temperature and solar radiation across all seasons, below average maximum temperature in the spring, and above average solar radiation in the winter–were each directly associated with a decrease in county-season averaged COVID-19 hospital admissions.

This study has implications for the potential mechanisms by which weather impacts respiratory infections, and how we build models of respiratory disease systems. In mathematical models of infectious disease transmission, the spread of infections from infected to susceptible individuals is defined as the product of the contact rate between susceptible and infected individuals, and the transmission probability if there is a contact. The common theory that weather impacts respiratory illnesses through changes in human behavior implies that weather alters contact rates. Our analysis suggests that, in the first year of the COVID-

19 pandemic, weather instead impacted the spread of SARS-CoV-2 by altering transmission probabilities. Below, we discuss the plausibility of this finding, as well as some caveats in the interpretation of this analysis.

One of the paradoxes in explaining the direct impacts of weather on transmission is explaining how outdoor temperatures directly impact the probability of transmission of a pathogen that is primarily spread in indoor environments. There is a body of evidence linking SARS-CoV-2 survival to ambient conditions, particularly evidence that SARS-CoV-2 survives longer at cooler temperatures and in drier conditions and that UV radiation accelerates virus decay [6,7,58]. However, we know that people spend most of their time indoors and that SARS-CoV-2 spreads most effectively in poorly ventilated indoor environments. Our comparison of indoor and outdoor weather conditions showed both indoor temperature and absolute humidity were strongly correlated with outdoor temperature and absolute humidity in all but the winter season (Fig 4). These patterns are consistent with Colorado weather, which typically require internal climate controls during the cold winter season and may also reflect people's willingness to open windows and/or limit indoor climate controls when the outside weather is comfortable. This high degree of correlation between indoor and outdoor weather conditions for much of the year may help to explain the stronger than anticipated direct effects of weather in our mediation analyses. For example, our all-season regression models indicated that high minimum temperature was associated with a decrease in 12-day lagged COVID-19 hospitalizations, which align with the existing body of evidence on the temperature-dependence of SARS-CoV-2 and the negative impacts that higher temperatures can have on virus survival and transmission [7,33]. By contrast, we also found that when absolute humidity was low during the winter season, there was a significant decrease in COVID-19 hospitalizations. This runs counter to the findings of other studies which have suggested that cold, dry conditions are likely to promote COVID-19 transmission, particularly in winter and in drier climates [59]. Given the low levels of correlation between indoor and outdoor absolute humidity during the winter in our study (see Figs 3 and 4), our findings may suggest a mitigating effect of indoor climate controls.

Our analysis, and the work of others, has found evidence that weather may impact mobility patterns in ways that act counter to observed relationships between weather and SARS-CoV-2 transmission. For example, we found the direct effects of high minimum temperature and high solar radiation was a significant reduction in 12-day lagged COVID-19 hospitalizations, whereas their indirect effects via time spent indoors away-from-home was a small increase in 12-day lagged COVID-19 hospitalizations. This aligns with the findings of similar studies conducted early in the pandemic, which reported a negative association between temperature/ solar radiation and COVID-19, but a positive association between temperature/solar radiation and human mobility concluding, like our study, that the impact of weather-driven mobility on COVID-19 may be counter to the direct effects of weather on mobility [18,19,60]. However, whereas these prior studies indicated that the strength of the indirect effects may be enough to offset any direct effects of weather, we found that the strength of the direct effects of weather far outweighed any suppression effects via human mobility. Another notable difference in our study was that our linear regression models also suggested that higher temperatures/solar radiation were associated with a significant decrease in time spent indoors away-from-home. This suggest that the effect of high minimum temperature/solar radiation was a reduction in time spent indoors away-from-home, and a subsequent increase in lagged COVID-19 hospitalizations, which runs counter to our hypothesis that reducing time spent indoors away-from-home would be protective against COVID-19 hospitalization risk.

An alternative explanation for our findings is that our measure of time spent indoors away-from-home did not accurately capture time spent indoors versus out, leading to

misclassification. We relied on Colorado building footprints from Microsoft (https://github.com/microsoft/USBuildingFootprints) to classify an individual as indoors or outdoors. The majority of the imagery used to build footprints for our study area was from 2019–2020, though footprints were also generated from older images when 2019–2020 data was not available [61], creating the possibility of misclassification bias, particularly in areas that have undergone rapid development in recent years. A comparison of our data to that of the United States' National Human Activity Pattern Survey (NHAPS) highlights that our study population spent significantly less time indoors during our study period than the average American in the 1990s: in our study we estimated that people spent 53.8% of their time indoors, whereas NHAPS estimated that Americans spent 87% of time spent in enclosed buildings [9]. While some of this difference is likely related to COVID-related stay-at-home mandates, we expected to see a larger uptick in time spent indoors away-from-home between spring and fall of 2020 – which shifted from 4.3% in the spring, to 5.6% in the fall–as schools and workplaces began resuming pre-pandemic operations.

To address this potential limitation, we reran the mediation analysis with time at home as the primary mediator. This measure was estimated based on the most common nighttime location and did not rely on building footprints, eliminating this potential source of misclassification. Our findings were consistent with the primary analysis, highlighting a strong direct effect of weather on COVID-19 hospitalizations, and a minor, if any, indirect effect of weather via changes in time at home. This suggests that our results are robust to our measure of mobility.

A key limitation of our study is the generalizability of our findings to transmission of SARS-CoV-2 today. We ended our study on January 31, 2021 because our access to the mobility data ended, and because of concerns about changing transmission dynamics due to the growing availability of vaccinations and the rise of highly transmissible SARS-CoV-2 variants in the late winter of 2021. During the first year of the pandemic, population immunity was very low and there were large alterations in human contact patterns due to widespread use of non-pharmaceutical interventions. This provided a unique opportunity to evaluate the mechanisms by which weather impacts an emerging pathogen. More research is needed to understand how these findings apply to more recent transmission dynamics of SARS-CoV-2 and other respiratory pathogens, as population immunity and contact patterns are distinct from our period of study. Nevertheless, the fact that the any direct weather or indirect mobility effects were discernible in this analysis in spite of the upheaval of mobility patterns suggests there is reason to investigate the generalizability of our findings to later phases of SAR-CoV-2 transmission.

Additional limitations include the ecological nature of our study. It is possible that individuals that were most likely to alter their weather-related mobility patterns were also those that had a lower risk of COVID-19 hospitalization, a phenomenon we were unable to investigate with our data. However, our results suggest that weather-related mobility changes–whether they are the result of more or less time indoors, outdoors, at home or away–had a small impact on COVID-19 hospitalizations, and were ultimately less influential than the direct impacts of weather. Given the relatively short study period, our analysis may have been underpowered to detect effect modification by season. Notably, only our all-season mediation models showed marginally statistically significant (p-value < 0.2) indirect effects, which may be a reflection of limited power for our season-stratified models.

## Conclusions

In this study, we set out to evaluate the extent to which weather impacts SARS-CoV-2 transmission via changes in mobility patterns, as measured using cell-phone data and building

footprints. We found evidence that weather, including temperature, absolute humidity and solar radiation directly impacted COVID-19 hospitalizations during the first year of the pandemic, and that the indirect mediation effect in which weather impacts mobility, and subsequently influences the likelihood of infection and hospitalization was either modest or null, in contrast to the direct effect. While other recent analyses have also highlighted similar direct effects of weather on SARS-CoV-2 and evidence of a suppression effect of weather via human mobility [18,19,60], our results indicated that the strength of the direct effects of weather on COVID-19 hospitalizations far outweighed any suppression effects. This has important implications, as it suggests weather may impact SARS-CoV-2 by altering transmission probabilities rather than contact patterns, and therefore measures that reduce the probability of transmission, via, for example vaccination or improved ventilation may be needed to reduce weather-driven changes in the spread of SARS-CoV-2. We recommend further analysis of this phenomenon to determine whether these findings generalize to current SARS-CoV-2 transmission dynamics wherein population immunity is high and human socialization and mobility patterns have largely returned to pre-pandemic levels.

## Supporting information

**S1 Table. Detailed regression results between control variables and categorical weather variables on time indoors away-from-home, and 12-day lagged hospitalizations.** (DOCX)

**S2 Table. Sensitivity analysis results of the linear regression results using categorical weather conditions on time at home and COVID hospitalizations.** (DOCX)

**S3 Table. Sensitivity analysis detailing mediation results of categorical weather conditions and time at home as the mediator.** (DOCX)

**S4 Table. Sensitivity analysis detailing linear regression results for continuous weather conditions on time indoors and away-from-home and COVID hospitalizations.** (DOCX)

**S5 Table. Sensitivity analysis results for the mediation models using continuous weather variables and time indoors away-from-home as the mediator.** (DOCX)

**S6 Table. Sensitivity analysis detailing linear regression results of continuous weather conditions on time at home, and on hospitalizations.** (DOCX)

**S7 Table. Sensitivity analysis detailing mediation results assessing time at home as a mediator between continuous weather conditions and COVID hospital admissions.** (DOCX)

**S8 Table. Sensitivity analysis detailing mediation results without hospitalization growth, using categorical weather conditions and time indoors away from home as the mediator.** (DOCX)

**S9 Table. Sensitivity analysis detailing mediation results without hospitalization growth, using categorical weather conditions and time at home as the mediator.** (DOCX)

## Acknowledgments

The authors would like to thank Kris Karnauskas for his help with developing appropriate measures of daily weather estimates for use in this analysis. Thank you to Jude Bayham for providing methodological and analytical feedback throughout this work. Thanks are also due to Gal Koss and David Johnson for cleaning and compiling the PurpleAir and X-mode data for use in this assessment, as we as to Brittney Contreras for designing Figures for this article.

## Author Contributions

**Conceptualization:** Elise N. Grover, Andrea G. Buchwald, Elizabeth J. Carlton.

**Data curation:** Elise N. Grover, Elizabeth J. Carlton.

**Formal analysis:** Elise N. Grover.

**Funding acquisition:** Elise N. Grover, Andrea G. Buchwald, Elizabeth J. Carlton.

**Methodology:** Elise N. Grover, Andrea G. Buchwald, Debashis Ghosh, Elizabeth J. Carlton.

**Project administration:** Elise N. Grover, Elizabeth J. Carlton.

**Resources:** Elizabeth J. Carlton.

**Supervision:** Elizabeth J. Carlton.

**Validation:** Debashis Ghosh.

**Visualization:** Elise N. Grover, Andrea G. Buchwald, Elizabeth J. Carlton.

**Writing – original draft:** Elise N. Grover, Andrea G. Buchwald, Elizabeth J. Carlton.

**Writing – review & editing:** Elise N. Grover, Andrea G. Buchwald, Debashis Ghosh, Elizabeth J. Carlton.

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
