## [Decision Letter · Decision Letter 0]

29 May 2024

Does behavior mediate the effect of weather on SARS-CoV-2 transmission? Evidence from cell-phone data

PONE-D-24-16434

Dear Dr. Carlton,

We’re pleased to inform you that your manuscript has been judged scientifically suitable for publication and will be formally accepted for publication once it meets all outstanding technical requirements.

Kind regards,

Rajeev Singh

Academic Editor

PLOS ONE

Additional Editor Comments (optional):

Reviewers' comments:

Reviewer's Responses to Questions

**Comments to the Author**

1. Is the manuscript technically sound, and do the data support the conclusions?

Reviewer #1: Yes

2. Has the statistical analysis been performed appropriately and rigorously? 

Reviewer #1: Yes

3. Have the authors made all data underlying the findings in their manuscript fully available?

Reviewer #1: Yes

4. Is the manuscript presented in an intelligible fashion and written in standard English?

Reviewer #1: Yes

5. Review Comments to the Author

Reviewer #1: This is an excellent analysis and is an important contribution to the debate that was raging early in the pandemic. I think it also offers some insights into other "seasonal" respiratory pathogens. I share this, not a requirement for publication, but as a suggestion in the event further work is done. One factor not considered in this analysis is age. While I have not looked at outside sources for these counties, I can say with certainty that hospitalization was skewed to the elderly with peculiar absence of young patients which signals a departure from some of the logic of seasonal mobility theorization. I share this as something to consider with so many cases coming from older populations which were not mobile but received seeding case visitors in their settings. It is a very robust analysis and I recommend its publication in PLOS ONE.

6. PLOS authors have the option to publish the peer review history of their article (what does this mean?). If published, this will include your full peer review and any attached files.

Reviewer #1: No

---

## [Editor Report · Acceptance letter]

7 Jun 2024

PONE-D-24-16434 

PLOS ONE

Dear Dr. Carlton, 

I'm pleased to inform you that your manuscript has been deemed suitable for publication in PLOS ONE. Congratulations! Your manuscript is now being handed over to our production team.

Kind regards, 

on behalf of

Dr. Rajeev Singh 

Academic Editor

PLOS ONE